# ReTAG: A Retrieved Cellular Topologies-Augmented Graph Learning Framework

## Abstract

Retrieval-augmented generation (RAG) methods for graph learning enhance the generalization of Graph Neural Networks (GNNs) by retrieving and integrating structurally relevant subgraphs, addressing their limitations on unseen or distribution-shifted graphs. However, current RAG-based methods mainly operate on zero- (nodes) and one-dimensional (edges) elements, failing to capture higher-dimensional topological structures, such as cycles, that are essential for identifying critical substructures and modeling complex relational patterns. This limitation hinders the retrieval of high-dimensional topological characteristics and weakens reasoning over graphs with complex higher-dimensional interactions. In this paper, we propose a novel **Re**trieved Cellular **T**opologies-**A**ugmented **G**raph Learning Framework (**ReTAG**), that leverages cellular complexes to model and retrieve multi-dimensional topology-aware subgraphs, termed cellular topologies. These structures encode multi-dimensional topological interactions across nodes, edges, and higher-dimensional cells. During inference, ReTAG retrieves cellular topologies based on their topological and semantic alignment with the input graph, and integrates them via a multi-dimensional topological message-passing mechanism that enables effective propagation of topological information across dimensions. Experiments on node classification, link prediction, and graph classification show ReTAG outperforms existing methods. The implementation code is available in the supplementary material.

## 1 Introduction

Graph representation learning encodes graph-structured data into low-dimensional embeddings that capture topological and semantic information, underpinning relational modeling in domains such as social networks Matsugu et al. (2023); Mane et al. (2025); Huang et al. (2025), biochemistry Yang et al. (2023); Xu et al. (2023), and traffic systems Zhang et al. (2025); Fang et al. (2025b). Within this context, Graph Neural Networks (GNNs) Kipf & Welling (2016b); Zhang et al. (2024), leveraging message-passing architectures, have become the dominant approach, enabling more expressive relational modeling than traditional node embedding methods Grover & Leskovec (2016). However, despite their effectiveness, GNNs often exhibit limited generalization to unseen graphs with substantially different topologies Zhao et al. (2024), a sensitivity that poses significant challenges for real-world applications where graph structures vary widely.

To overcome this limitation, retrieval-augmented methods for graph learning have recently emerged as a promising approach Lewis et al. (2021); Jiang et al. (2024), drawing inspiration from retrieval-augmented generation (RAG) techniques in NLP. RAG methods enhance GNNs by retrieving external subgraphs that are structurally or semantically relevant to the input, and integrating them into the learning pipeline to provide additional contextual information. Notably, RAGRAPH Jiang et al. (2024) have demonstrated enhanced adaptability by incorporating retrieved graph fragments through message-passing prompt mechanisms. However, existing RAG methods focus mainly on low-dimensional elements (nodes and edges), overlooking higher-dimensional structures like cycles that are essential for retrieving meaningful substructures and reasoning over complex relations.

In many real-world graphs (as illustrated in Fig.1), critical structural information arises not only from 0-dimensional nodes and 1-dimensional edges, but also from high-dimensional

motifs—cycles—that collectively form cell complexes Bodnar et al. (2021b), a foundational notion in algebraic topology Hatcher (2000). High-dimensional topological structures, like cycles in molecular graphs, capture domain-specific inductive biases vital for accurate interpretation and retrieval. However, conventional GNN-RAG (Fig.1(a)) methods struggle to retrieve meaningful analogues when they fail to recognize or reason over such high-dimensional structures. In contrast, retrieving cyclic motifs from external graph corpora (Fig.1(b)) can provide structurally analogous examples that support inference in novel molecules, especially when query graphs exhibit global patterns that differ significantly from those during training. Moreover, high-dimensional reasoning captures complex dependencies beyond traditional pairwise message passing.

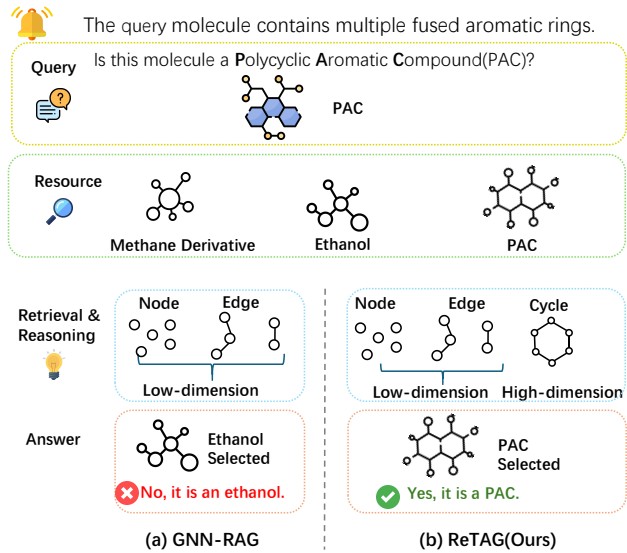

Figure 1: RAG-based graph learning with multi-dimensional topological characteristics.

In this work, we propose a novel **Re**trieved Cellular **T**opologies-**A**ugmented **G**raph Learning Framework (**ReTAG**) that leverages cellular complexes to model and retrieve multi-dimensional topology-aware subgraphs, termed cellular topologies. Specifically, ReTAG first lifts input graphs into cellular complexes to capture high-dimensional topological structures such as cycles, enabling the extraction of multi-dimensional topology-aware substructures. These substructures are organized into a knowledge base that encodes rich topological and semantic information beyond traditional nodes and edges. During inference, ReTAG retrieves cellular topologies that align both topologically and semantically with the input graph, and integrates them through a multi-dimensional topological message-passing mechanism designed to propagate topological information effectively across different dimensions. Furthermore, a cellular topological contrastive learning module is introduced to enhance structural discrimination by enforcing feature consistency within each 2-cell, thereby capturing topological semantics beyond conventional pairwise interactions. Extensive experiments demonstrate that ReTAG significantly outperforms existing methods across diverse graph scenarios.

## 2 RELATED WORKS

Graph representation learning aims to capture structural and semantic information from graph-structured data. GNNs have become a dominant tool in this area by aggregating information through message passing Cai et al. (2018); Liu et al. (2021b); Bodnar et al. (2021b;d). Despite their success, GNNs struggle with dynamic or multi-task settings due to rigid architectures and fixed message-passing schemes that lack task-aware modulation and flexible inference control.

To improve model flexibility and task adaptivity, recent studies have proposed graph prompt learning, inspired by prompt-based tuning in natural language processing Wei et al. (2023); Zhou et al. (2023). This paradigm typically involves pre-training GNNs on large-scale graph Hu et al. (2020); You et al. (2020a); Qiu et al. (2020); Zhao et al. (2024); Yu et al. (2024) and guiding downstream task execution through prompt-like structures that inject task-relevant information. For example, VNT Tan et al. (2023) introduces virtual nodes as soft prompts to encode task-level context and guide message propagation. GraphPrompt Liu et al. (2023b) proposes a task-specific readout mechanism that learns to aggregate node representations differently depending on the downstream objective. GraphPro Yang et al. (2024b) further designs spatial- and temporal-aware gating mechanisms, adapting the prompt to dynamic graph settings like recommender systems. PRODIGY Huang et al. (2023b) constructs a task graph as an explicit prompt structure and learns cross-graph in-context learning capabilities by interacting it with a data graph via a transformer-based module.

ProNoG Yu et al. (2025a) proposes a novel pre-training and prompt learning framework tailored for non-homophilic graphs, introducing a conditional network that captures node-specific relational patterns to enhance downstream task performance. Graph prompt learning typically presumes structural alignment with pretraining data, hindering generalization to graphs with divergent topologies.

To overcome this limitation, recent work has drawn inspiration from RAG in NLP Lewis et al. (2021); Zhao et al. (2023), where external information is retrieved and injected into a pre-trained model to support better generalization. In the context of graph learning, a representative effort is RAGRAPH Jiang et al. (2024), which enhances GNNs through a retrieval-based framework that leverages a toy-graph library to incorporate structurally or semantically similar graphs via message-passing prompting, enabling adaptation without retraining. Despite their effectiveness, existing graph RAG methods primarily rely on low-dimensional structures, overlooking higher-dimensional topologies crucial for capturing key substructures and complex relations.

Additional related works on topological deep learning are discussed in Appendix C.

## 3 PRELIMINARIES

***Definition 1. (Cell Complex Hansen & Ghrist (2019)).*** A **regular cell complex** is a topological space $X$ decomposed into a collection of disjoint subspaces $\{x_\alpha\}_{\alpha \in P_X}$, referred to as *cells*, satisfying the following conditions:

1. For each point $p \in X$, there exists an open neighborhood intersecting only finitely many cells.

2. For any pair of cells $x_\alpha, x_\tau$, the intersection $x_\tau \cap \overline{x_\alpha}$ is nonempty if and only if $x_\tau \subseteq \overline{x_\alpha}$, where $\overline{x_\alpha}$ denotes the topological closure of $x_\alpha$.

3. Each cell $x_\alpha$ is homeomorphic to an open ball in $\mathbb{R}^n$ for some non-negative integer $n$.

4. *(Regularity)* The closure $\overline{x_\alpha}$ of every cell is homeomorphic to a closed ball in $\mathbb{R}^{n_\alpha}$, with the interior mapped homeomorphically onto $x_\alpha$ itself.

***Definition 2.*** A **cellular lifting map** is a function $f : \mathcal{G} \to X$ from the space of graphs $\mathcal{G}$ to the space of regular cell complexes $X$, satisfying that two graphs $G_1, G_2 \in \mathcal{G}$ are isomorphic if and only if their corresponding cell complexes $f(G_1)$ and $f(G_2)$ are isomorphic. Intuitively, a cell complex is built hierarchically by first considering 0-cells (vertices), then attaching 1-cells (edges) via their endpoints, and further incorporating higher-dimensional cells—such as 2-cells—by gluing disks along cycles.

***Definition 3. (Retrieved Cellular Topologies-Augmented Graph Learning).*** Given a graph $\mathcal{G}$, a cellular lifting map $f : \mathcal{G} \to X$ is conducted from the space of graphs to the space of regular cell complexes. To construct a cell complex knowledge base $X = \{X_\sigma\}$, we extract localized subcomplexes from the evolving complex $X$ centered around a master 0-cell $x_m^0$, defined as the 0-cell with the highest degree in the complex. Each subcomplex includes the set k-hop$_{X^{(0)} \cup X^{(1)}}(x_m^0)$ of 0- and 1-cells reachable within $k$ hops from $x_m^0$, as well as all 2-cells $x^2 \in X^{(2)}$ such that there exists a cell $x^1 \in$ k-hop$_{X^{(0)} \cup X^{(1)}}(x_m^0)$ with $x^1 \prec x^2$, meaning $x^1$ is a face of $x^2$. To enable retrieval, each cell complex $X_\sigma$ is indexed by a composite **key** that includes: the timestamp $\tau_b$, the master 0-cell embedding $h_m^0$, and a two-dimensional topological characteristic $h_\sigma^2$ obtained by pooling over all 2-cells within $X_\sigma$. Retrieval is performed via similarity over keys to obtain **values** such as task-specific output vectors $\{o_i \in \mathbb{R} \mid x_i \in X_\sigma\}$ and cell complex embeddings $\{h_i \in \mathbb{R} \mid x_i \in X_\sigma\}$.

Given a graph $\mathcal{G}$, we first split it into training and testing subsets, $\mathcal{G} = \mathcal{G}_{\text{train}} \cup \mathcal{G}_{\text{test}}$, and lift them into a regular cell complex $X = X_{\text{train}} \cup X_{\text{test}}$ via a map $f : \mathcal{G} \to X$. Each unit (e.g., a 0-cell $x_i^0$, 1-cell $x_{ij}^1$, or complex $X_\sigma$) has label $y_i$ if and only if it resides in $X_{\text{train}}$. The goal is to predict $Y_{\text{test}}$ by retrieving relevant cell complexes and propagating structure-aware representations.

## 4 THE RETAG FRAMEWORK

In this section, we present the architecture of ReTAG (illustrated in Figure 2), a retrieval-augmented graph learning framework built upon multi-dimensional cellular topologies. ReTAG is composed of three key components: a *Cellular Representation and Knowledge Extraction* module that lifts

Figure 2: The overview of Retrieval Cellular Topologies-Augmented Graph Learning Framework.

input graphs into cellular complexes and constructs a cell complex knowledge base; a *Retrieval-Augmented Graph Inference* module that retrieves semantically and topologically aligned subcomplexes to guide graph learning; and a *Cellular Topological Contrastive Learning* module that regularizes learning and enhances generalization through topology-aware objectives.

## 4.1 CELLULAR REPRESENTATION AND KNOWLEDGE EXTRACTION

**Fundamental Cycle-Guided Complex Lifting.** We construct a 2-dimensional regular cell complex from the task graph $G = (V, E, T)$ by first treating it as a 1-dimensional Cell complex, where: Each vertex $v \in V$ is a 0-cell, forming the 0-skeleton $X^{(0)}$; Each edge $(u, v) \in E$ is a 1-cell attached to the 0-skeleton, yielding the 1-skeleton:

$$X^{(1)} = X^{(0)} \cup \{ e \mid (u, v) \in E \}. \tag{1}$$

To identify higher-dimensional structures, we first fix a spanning tree $\mathcal{T} \subseteq G$. Since $\mathcal{T}$ is contractible, we can collapse it to a single point via the quotient map:

$$\gamma : G \to G/\mathcal{T}, \tag{2}$$

which collapses the entire subtree $\mathcal{T}$ to a single point. Here, $G/\mathcal{T}$ denotes the quotient space obtained by identifying all vertices and edges of $\mathcal{T}$ to a single point.

Under this contraction, each edge $e = (u, v) \in E \setminus \mathcal{T}$ corresponds to a fundamental cycle formed by adding $e$ to the collapsed tree point, as the non-tree edges become loops attached to that point. The preimage $\gamma^{-1}(e)$ recovers the unique cycle $c_e$ in the original graph, consisting of $e$ together with the tree path in $\mathcal{T}$. For each such cycle, we attach a 2-cell by choosing an attaching map:

$$\varphi_e : \partial D^2 \cong S^1 \longrightarrow c_e \subset X^{(1)}, \tag{3}$$

and extending it via a characteristic map

$$\Phi_e : D^2 \longrightarrow X, \quad \Phi_e|_{\partial D^2} = \varphi_e,$$

where the attaching map $\varphi_\alpha$ glues the boundary of the 2-cell $D^2$ onto the 1-skeleton $X^{(1)}$, $x_e^2 = \Phi_e(D^2)$ defines the 2-cell corresponding to edge $e$.

Formally, the collection of these 2-cells constitutes

$$X^{(2)} = \{x_e^2 \simeq D^2 \mid e = (u, v) \in E \setminus \mathcal{T}\}, \tag{4}$$

where each disk $D^2$ is attached along its boundary $\partial D^2 \cong S^1$ via the loop $c_e \subset X^{(1)}$.

**Proposition 1.** *(Proof in Appendix D.1.) $G/\mathcal{T}$ is homotopy-equivalent to $G$, and $\gamma$ induces an isomorphism on the first homology group $H_1(G; \mathbb{Z})$.*

**Proposition 2.** *(Proof in Appendix D.2.) Each non-tree edge $e \in E \setminus \mathcal{T}$ induces a unique fundamental cycle in $G$, which becomes a nontrivial loop in $G/\mathcal{T}$. The collection of these loops forms a basis of the first homology group $H_1(G; \mathbb{Z})$, capturing all independent cycles and providing a concise topological summary of the graph.*

**Multi-dimensional Topological Message Passing (MTMP).** We design a two-stage message passing mechanism leveraging the hierarchical structure of the cell complex. The first stage propagates information along the 1-skeleton via alternating message exchanges between nodes (0-cells) and edges (1-cells) over $L$ hops. The second stage allows cells of all dimensions to propagate and aggregate high-dimensional information via their attached higher-dimensional cells (2-cells), enabling rich topological context exchange across different cell dimensions.

To model these interactions, we define three message passing operations over each cell complex $x$, corresponding to its faces, cofaces, and adjacent complexes:

$$
\begin{aligned}
m_{\mathcal{F}}^{l+1}(x) &= \text{AGG}_{w \in \mathcal{F}(x)} \left( M_{\mathcal{F}} \left( \boldsymbol{h}_x^l, \boldsymbol{h}_w^l \right) \right), \\
m_{\mathcal{C}}^{l+1}(x) &= \text{AGG}_{w \in \mathcal{C}(x)} \left( M_{\mathcal{C}} \left( \boldsymbol{h}_x^l, \boldsymbol{h}_w^l \right) \right), \\
m_{\uparrow}^{l+1}(x) &= \text{AGG}_{w \in \mathcal{N}_{\uparrow}(x)} \left( M_{\uparrow} \left( \boldsymbol{h}_x^l, \boldsymbol{h}_w^l, \boldsymbol{h}_{x \cup w}^l \right) \right),
\end{aligned} \tag{5}
$$

where $l$ indicates the iteration step, $\mathcal{F}(x) = \{y \mid y \prec x\}$ and $\mathcal{C}(x) = \{z \mid x \prec z\}$ denote the sets of faces and cofaces of complex $x$, respectively, with $\prec$ representing the face relation (i.e., $y \prec x$ means $y$ is a face of $x$). The set $\mathcal{N}_{\uparrow}(x)$ contains complexes adjacent to $x$ via a shared coface.

We incorporate the above message types into a two-stage message propagation scheme: The first stage enables a topological expansion over 1-skeleton between 0-cell and 1-cell complexes. The representation is updated as:

$$
\boldsymbol{h}_x^l = \text{UPDATE} \left( \boldsymbol{h}_x^l, m_{\mathcal{F}}^l(x), m_{\mathcal{C}}^l(x) \right), \tag{6}
$$

where the first-stage message passing iterates up to $L$ hops.

The second stage integrates the three message types, allowing cells of all dimensions to propagate and aggregate high-dimensional information through their attached 2-cells, thereby enriching the representation as:

$$
\boldsymbol{h}_x^{(L+1)} = \text{UPDATE} \left( \boldsymbol{h}_x^L, m_{\mathcal{F}}^L(x), m_{\mathcal{C}}^L(x), m_{\uparrow}^{L+1}(x) \right). \tag{7}
$$

**Cell Complex Knowledge Base Construction.** Finally, we construct the cell complex knowledge base $X = \{X_{\sigma}\}$. To mitigate the dominance of high-degree 0-cells (i.e., nodes) and better capture long-tail topological patterns, we adopt an inverse importance sampling strategy for selecting seed 0-cell complexes. The importance score of each 0-cell complex $x$ is defined as a convex combination of its PageRank score $\text{PR}(x)$ and its degree $\deg(x)$ using the formula $I(x) = \alpha \cdot \text{PR}(x) + (1 - \alpha) \cdot \deg(x)$, where $\alpha \in [0, 1]$ balances the relative weight between centrality and connectivity. Sampling probabilities are computed by normalized inverse importance:

$$
p(x_i) = \frac{1/(I(x_i) + \varepsilon)}{\sum_j 1/(I(x_j) + \varepsilon)}, \tag{8}
$$

where $\varepsilon > 0$ is a small constant ensuring numerical stability.

Each local cell complex $\mathcal{X}_{\sigma}$ is defined as the $k$-hop neighborhood around the master 0-cell $x_m^0$, i.e., the 0-cell with the highest degree in the complex. It comprises all 0-cells and 1-cells reachable within $k$ hops from $x_m^0$, along with all 2-cells having at least one 1-cell face within this neighborhood. To enhance structural diversity while preserving higher-dimensional topology, we apply topology-aware augmentations—such as node dropout, edge rewiring, and Gaussian perturbation—under the constraint that no 2-cell is disrupted. These augmentations generate multiple structurally consistent instances of the local complex, facilitating robust learning of topological representations.

For each local complex $X_{\sigma}$, the key is defined as $\boldsymbol{k}_b = [\tau_b, \boldsymbol{h}_m^0, \boldsymbol{h}_{\sigma}^2]$, where $\tau_b$ is the current time step, $\boldsymbol{h}_m^0$ denotes the embedding of the master 0-cell, and $\boldsymbol{h}_{\sigma}^2$ is a pooled two-dimensional topological feature summarizing all 2-cells within $X_{\sigma}$. The corresponding value is $\boldsymbol{v}_b = \{[\boldsymbol{o}_i, \boldsymbol{h}_i] \mid x_i \in X_{\sigma}\}$, containing the output vectors $\boldsymbol{o}_i$ and embeddings $\boldsymbol{h}_i$ of the constituent cell complexes.

## 4.2 RETRIEVAL-AUGMENTED GRAPH INFERENCE WITH TOPOLOGICAL COMPLEXES

**Cell Complex Retrieval Process.** Given a query graph $\mathcal{G}_q$, we first apply the lifting map $f : \mathcal{G} \to X$ and employ the Multi-dimensional Topological Message Passing framework to obtain its

cell complex representations. The query key is defined as $\boldsymbol{k}_q = [\tau_q, \boldsymbol{h}_c^0, \boldsymbol{h}_q^2]$, where $\tau_q$ denotes the time step, $\boldsymbol{h}_c^0$ is the embedding of the central 0-cell, and $\boldsymbol{h}_q^2$ represents a pooled two-dimensional topological feature summarizing all 2-cells within the query complex $X_q$. The $topK$ cell complexes $X_{topK}$ are selected based on a similarity score computed as a weighted combination of the factors:

$$S(\boldsymbol{k}_q, \boldsymbol{k}_b) = \mathbf{w} \times [S_\tau(\tau_q, \tau_b), S_0(\boldsymbol{h}_c^0, \boldsymbol{h}_m^0), S_2(\boldsymbol{h}_q^2, \boldsymbol{h}_\sigma^2)]^\mathsf{T}, \tag{9}$$

where $\mathbf{w} = [\mathbf{w}_1, \mathbf{w}_2, \mathbf{w}_3]$ denotes a set of hyperparameter weights, $S_\tau(\tau_q, \tau_b) = e^{-\eta|\tau_q - \tau_b|}$ captures temporal similarity with a decay rate $\eta$, and both $S_0(\cdot, \cdot)$ and $S_2(\cdot, \cdot)$ compute cosine similarity between the corresponding features.

**Cellular Knowledge Injection Propagation.** Within each retrieved cell complex $X_\sigma \in X_{\text{TopK}}$, we perform *intra-complex aggregation* to propagate information from the constituent cells to the master 0-cell using the proposed Multi-dimensional Topological Message Passing (MTMP) module. Specifically, both the task-specific output vectors $\boldsymbol{o}_i$ and hidden embeddings $\boldsymbol{h}_i$ of the constituent cells $x_i \in X_\sigma$ are aggregated and transmitted to the master 0-cell $x_m^0$:

$$\mathbf{z}_m^0 = \text{MTMP}\left(\{\mathbf{z}_i \mid x_i \in X_\sigma\}\right), \tag{10}$$

where $\mathbf{z}_i \in \{\boldsymbol{o}_i, \boldsymbol{h}_i\}$ and $\mathbf{z}_m^0 \in \{\boldsymbol{o}_m^0, \boldsymbol{h}_m^0\}$. For parameter-free settings, such propagation can be pre-computed and cached during knowledge base construction to improve efficiency during inference.

Then we perform *inter-complex aggregation*, where the retrieved master 0-cells $x_m^0$ from the selected cell complexes $X_\sigma \in X_{\text{TopK}}$ act as knowledge anchors to enhance the query center 0-cell $x_c^0$ in $X_q$. The MTMP module is reused to inject knowledge from each $x_m^0$ into $x_c^0$. Although the retrieval and connection occur at the 0-cell level, the injected information is further propagated to higher-dimensional cells through multi-level message passing over the entire cell complex. Formally, for each query center cell:

$$\boldsymbol{h} = \text{MTMP}\left(\{\boldsymbol{h}_i \mid x_i \in X_q \cup \{x_m^0\}\}\right), \tag{11}$$

$$\boldsymbol{o}_c = \text{MTMP}\left(\{\boldsymbol{o}_i^0 \mid x_i^0 \in \{x_m^0\}\}\right), \tag{12}$$

where $\boldsymbol{h} = [\{\boldsymbol{h}^0\}, \{\boldsymbol{h}^1\}, \{\boldsymbol{h}^2\}]$ concludes embedding of various dimensional cell complexes.

Finally, at the data fusion layer, the aggregated hidden state $\boldsymbol{h}_c^0$ of the center 0-cell $x_c^0$ and $\boldsymbol{h}_q^2$ the pooling of 2 cell complexes is passed through a multi-layer perceptron $\text{MLP}(\cdot)$, yielding a structure-aware output. This output is then combined with the task-specific output $\boldsymbol{o}_c$, which has already integrated multi-dimensional signals through the retrieval-enhanced process. The final prediction is computed as:

$$\hat{\mathbf{o}}_c = \gamma \boldsymbol{o}_c + (1 - \gamma)\text{MLP}([\boldsymbol{h}_c^0, \boldsymbol{h}_q^2]), \tag{13}$$

where $\gamma \in [0, 1]$ is a reweighting hyperparameter. The resulting representation $\hat{\mathbf{o}}_c$ serves as the input to downstream node-, edge-, or graph-level tasks, typically evaluated via a task-specific similarity function.

### 4.3 Cellular Topological Contrastive Learning (CTCL)

We propose **Cellular Topological Contrastive Learning (CTCL)** to encode high-dimensional structural semantics within each 2-cell $x_k^2 \in X^{(2)}$ by enforcing feature consistency among its constituent 0-cells. Formally, let $\mathcal{X}_k^{(0)}$ denote the set of 0-cells forming $x_k^2$. For each 2-cell, we independently sample two subsets of 0-cells, $\tilde{\mathcal{X}}_{k,1}^{(0)}$ and $\tilde{\mathcal{X}}_{k,2}^{(0)}$, using degree-based importance sampling. The embeddings of these subsets are aggregated to produce two representations per 2-cell:

$$\tilde{\boldsymbol{h}}_{k,v} = \frac{1}{|\tilde{\mathcal{X}}_{k,v}^{(0)}|} \sum_{x^0 \in \tilde{\mathcal{X}}_{k,v}^{(0)}} \boldsymbol{h}_{x^0} + \alpha \cdot \mathcal{N}(0,1), \quad k = 1, \ldots, N, ; v \in 1, 2, \tag{14}$$

where $\boldsymbol{h}_{x^0}$ is the embedding of 0-cell $x^0$, $\alpha$ controls the magnitude of Gaussian perturbation, and $N = |X^{(2)}|$ is the number of 2-cells. We adopt an InfoNCE loss to enforce consistency between these two views:

$$\mathcal{L}_{\text{CTCL}} = -\frac{1}{N} \sum_{k=1}^N \log \frac{\exp\left(\text{sim}(\tilde{\boldsymbol{h}}_{k,1}, \tilde{\boldsymbol{h}}_{k,2})/\tau\right)}{\sum_{j \neq k} \exp\left(\text{sim}(\tilde{\boldsymbol{h}}_{k,1}, \tilde{\boldsymbol{h}}_{j,2})/\tau\right)}, \tag{15}$$

where $\text{sim}(\cdot, \cdot)$ denotes the cosine similarity and $\tau$ is a temperature parameter. The overall training objective combines standard classification with topological contrastive regularization:

$$\mathcal{L}_{\text{cls}} = -\sum_{c=1}^{C} y_c \log \hat{\mathbf{y}}_c, \tag{16}$$

$$\mathcal{L}_{\text{total}} = \mathcal{L}_{\text{cls}} + \lambda \cdot \mathcal{L}_{\text{CTCL}}, \tag{17}$$

where $y_c$ and $\hat{\mathbf{y}}_c$ denote the ground-truth label and predicted probability for class $c$, respectively, and $\lambda$ balances supervised learning and topological contrastive regularization.

## 5 EXPERIMENTS

### 5.1 EXPERIMENTAL SETUP

**Datasets.** We evaluate our framework on three tasks: node classification, graph classification, and link prediction. Node classification is performed on PROTEINS and ENZYMES Morris et al. (2020), graph classification on ENZYMES, PROTEINS, COX2, and BZR Morris et al. (2020); Rossi & Ahmed (2015c), and link prediction on two dynamic recommendation datasets: KOUBEI Zhu et al. (2021) and AMAZON He & McAuley (2016). See Appendix E.1, Table 5 for details.

**Comparison Methods.** To verify the effectiveness of ReTAG, we conduct comprehensive experiments across various graph learning tasks. For dynamic graphs, we include comparisons with *GraphCL-based Methods*: LightGCN (He et al., 2020c), SGL (Wu et al., 2021b), MixGCF (Liu et al., 2021a), SimGCL (Yu et al., 2021), *Prompt-based*: GraphPro (Liu et al., 2023c), Graph-Pro+PRODIGY (Feng et al., 2023), and *Retrieval-Augmented*: RAGraph (Jiang et al., 2024). For static graphs, we compare ReTAG with representative models, including *GNN Baselines*: GCN (Kipf & Welling, 2017a), GraphSAGE (Hamilton et al., 2017b), GAT (Veličković et al., 2018), GIN (Xu et al., 2019b), *Retrieval-Augmented Variants*: RAGraph (Jiang et al., 2024), *Prompt-based Variants*: GraphPrompt (Xie et al., 2022), GraphPrompt+PRODIGY (Feng et al., 2023), and ProNoG (Yu et al., 2025a).

**Settings and Evaluation.** We adopt a training–resource split: training is on labeled data, and retrieval is from a disjoint resource set. For static graphs, we use a 50%:30% node split Liu et al. (2023c); for dynamic graphs, snapshots are chronologically split for time-aware retrieval Huang et al. (2024). Retrieval-based methods (e.g., PRODIGY, RAGraph) retrieve only from the resource set to prevent information leakage, while other baselines are fine-tuned on the merged graph. We report accuracy for classification and Recall@20/nDCG@20 for link prediction. All experiments use PyTorch with an NVIDIA A100 GPU (40GB). We use Adam (lr=1e-3, batch size=16). For classification, the hidden dimension is set to 256, and the first stage of MTMP employs a single layer ($L = 2$). Cell complexes are built via $k$-hop neighborhoods, with $k$ selected by validation (typically 2 or 3). CTCL applies a 2 dimensional contrastive loss with weight $\lambda \in [0.1, 0.4]$. In the link prediction task, the first stage of MTMP adopts a 3-layer architecture ($L = 3$) with 64-dimensional embeddings, and the model is trained end-to-end using contrastive and retrieval modules.

### 5.2 CELLULAR COMPLEX-AUGMENTED GRAPH RESULTS

**Overall Performance.** The experimental results of our model and baselines on node and graph classification for static graphs and on link prediction for dynamic graphs are summarized in Table 1 and Table 2. Based on the comparison with existing baselines, we draw the following observations:

- **Our proposed ReTAG achieves the best performance.** ReTAG significantly outperforms all baselines on both static and dynamic graphs. On static datasets, ReTAG achieves a relative improvement of 2.3%–3.7% in node classification accuracy and 1.9%–4.4% in graph classification accuracy over state-of-the-art methods, consistently achieving the highest performance across all tasks. On dynamic benchmarks, ReTAG achieves relative gains of 2.7%–6.7% in Recall@20 and 2.3%-6.2% in nDCG@20 over strong baselines such as RAGraph/FT. These gains can be attributed to ReTAG's ability to incorporate high-dimensional topological knowledge via cell complex lifting and retrieve structured support from external graphs. Its MTMP module enables hierarchical propagation across complex structures, while the CTCL loss regularizes representations by enforcing 2-cell consistency, jointly enhancing expressiveness and generalizability across tasks.

Table 1: Accuracy evaluation on node and graph classification. All tabular results (%) are reported as mean±std across five runs. The best results are **bolded** and the runner-ups are underlined.

| Methods | Node Classification | | Graph Classification | | | |
|---|---|---|---|---|---|---|
| | PROTEINS (5-shot) | ENZYMES (5-shot) | PROTEINS (5-shot) | COX2 (5-shot) | BZR (5-shot) | ENZYMES (5-shot) |
| *GNN Baselines* | | | | | | |
| GCN | 46.63±03.04 | 52.80±12.89 | 54.80±06.64 | 67.87±03.39 | 58.76±05.08 | 22.67±05.20 |
| GraphSAGE | 48.87±02.64 | 48.75±01.59 | 54.68±09.34 | 67.02±05.42 | 58.27±04.79 | 21.17±05.49 |
| GAT | 48.13±07.90 | 47.75±01.23 | 55.82±07.31 | 64.89±03.23 | 57.04±06.70 | 20.67±03.27 |
| GIN | 49.61±01.58 | 48.82±01.58 | 56.17±08.58 | 62.77±02.85 | 56.54±04.20 | 21.10±02.53 |
| *GraphPrompt+ Variants* | | | | | | |
| Vanilla/NF | 44.88±13.17 | 48.81±01.88 | 56.68±03.63 | 53.04±04.13 | 68.77±03.44 | 36.50±03.31 |
| Vanilla/FT | 48.99±01.88 | 51.99±01.36 | 57.04±03.88 | 64.04±08.20 | 69.01±02.21 | 40.00±04.36 |
| PRODIGY/NF | 47.32±08.12 | 43.80±14.03 | 53.48±06.72 | 53.97±10.34 | 67.18±08.93 | 22.12±13.84 |
| PRODIGY/FT | 53.26±06.42 | 57.98±12.37 | 57.14±10.34 | 65.31±04.28 | 68.08±06.68 | 25.94±05.12 |
| ProNoG | 52.89±11.76 | 76.48±18.23 | 61.63±08.01 | 58.03±14.28 | 62.26±12.27 | 37.91±05.64 |
| *Retrieval-Augmented Variants* | | | | | | |
| RAGraph/NF | 56.12±04.11 | 75.92±01.72 | 58.48±03.93 | 55.32±04.15 | 77.53±05.26 | 38.17±03.39 |
| RAGraph/FT | 58.74±00.87 | 75.74±01.92 | 62.33±02.52 | 76.60±02.30 | 76.79±05.02 | 47.71±06.88 |
| **ReTAG(Ours)** | **60.91**±01.79 | **78.26**±01.57 | **64.30**±02.21 | **78.09**±02.57 | **79.01**±03.49 | **49.83**±04.72 |

Table 2: Performance evaluation (%) on link prediction.

| Methods | KOUBEI | | AMAZON | |
|---|---|---|---|---|
| | Recall@20 | nDCG@20 | Recall@20 | nDCG@20 |
| *GraphCL* | | | | |
| LightGCN | 30.21±06.45 | 22.24±05.83 | 15.07±06.48 | 06.53±02.66 |
| SGL | 32.61±04.27 | 22.36±04.82 | 15.78±07.12 | 07.90±02.49 |
| MixGCF | 32.06±04.20 | 22.49±06.91 | 15.24±08.98 | 07.40±03.44 |
| SimGCL | 33.07±05.28 | 23.08±05.55 | 16.10±07.91 | 07.58±03.51 |
| *GraphPro+* | | | | |
| Vanilla/NF | 21.31±04.59 | 15.31±03.11 | 12.56±07.45 | 06.31±03.92 |
| Vanilla/FT | 33.96±04.13 | 24.66±02.78 | 18.14±07.55 | 08.73±03.74 |
| PRODIGY/NF | 21.66±03.21 | 14.82±03.92 | 11.88±02.61 | 05.84±01.84 |
| PRODIGY/FT | 33.46±04.70 | 23.28±03.40 | 16.72±04.28 | 08.09±02.66 |
| *Retrieval-Augmented Variants* | | | | |
| RAGRAPH/NF | 22.86±03.44 | 16.68±02.48 | 13.78±05.54 | 06.52±02.69 |
| RAGRAPH/FT | 34.27±03.93 | 24.82±02.69 | 18.32±07.45 | 09.09±03.89 |
| **ReTAG (Ours)** | **35.21**±03.21 | **25.39**±02.23 | **19.54**±08.04 | **09.65**±04.06 |

- **Retrieval-augmented topology reasoning is crucial for generalized graph learning.** From Table 1 and Table 2, we observe that retrieval-augmented baselines (e.g., RAGraph/FT) consistently outperform standard GNNs and prompt-based variants, demonstrating that incorporating external knowledge via retrieval is vital for generalization. However, these baselines often rely on shallow node- or edge-level similarity, limiting their ability to capture complex structural semantics. ReTAG addresses this limitation by lifting graphs into higher-dimensional cell complexes and conducting retrieval based on both semantic and topological alignment. This enables the model to obtain more informative and transferable support subgraphs. With MTMP injection and CTCL regularization, ReTAG better integrates retrieved knowledge, yielding superior performance on various graph tasks.

**Hyper-parameter Study.** We study the sensitivity of ReTAG to two key hyper-parameters: the number of hops ($k$) used to construct cell complexes, and the cellular topological contrastive learning (CTCL) loss weight. The $k$-hop value determines the receptive field of each complex, affecting the amount of structural context for retrieval and reasoning. The contrastive loss weight controls the strength of 2-cell topological regularization during training. We vary $k$ from 1 to 4, and the loss weight from 0 to 0.5 to assess their respective impacts. Figure 3 shows the effect of different $k$-hop values on classification accuracy. As $k$ increases, the volume of retrieved knowledge expands accordingly. While larger $k$ introduces noisy or redundant substructures that hinder performance, excessively small $k$ may miss crucial topological patterns needed for effective retrieval. Figure 4 illustrates the impact of varying the CTCL loss weight. Insufficient weight underutilizes the discriminative 2-dimensional information, while too large a weight overly constrains learning, interfering with the main optimization objective and degrading downstream performance. These

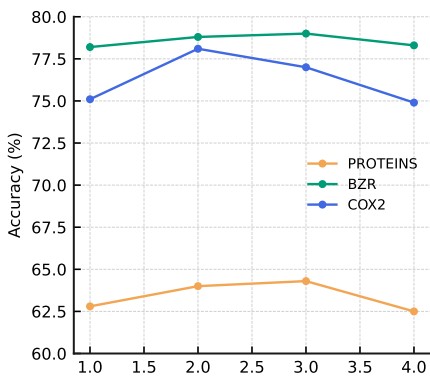 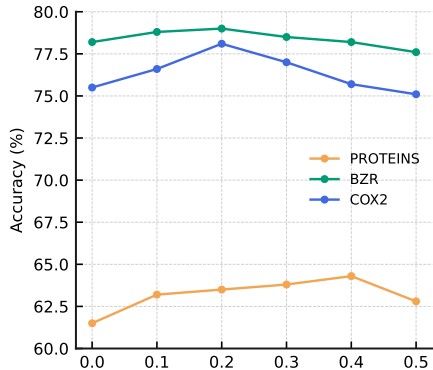

Figure 3: Accuracy with varying k-hop size.    Figure 4: Accuracy with varying CTCL weight.

trends highlight the importance of balancing topological regularization with task-specific learning in ReTAG.

**Ablation Study.** We evaluate three ablated variants of ReTAG: removing the Fundamental Cycle-Guided Complex Lifting (FCL), which eliminates 2-cell construction and reduces the model to an edge-based graph; discarding the Multi-dimensional Topological Message Passing (MTMP), which disables hierarchical propagation across cell dimensions; and removing the Cellular Topological Contrastive Learning (CTCL), which drops contrastive regularization over 2-cell representations. As shown in Table 3, we observe that:

Table 3: Ablation study on COX2 and PROTEINS datasets.

| Model Variant | COX2 (%) | PROTEINS (%) |
|---|---|---|
| w/o FCL | 76.05±02.89 | 62.33±02.52 |
| w/o MTMP | 76.21±01.97 | 62.45±03.19 |
| w/o CTCL | 76.94±01.83 | 63.21±04.79 |
| **Full Model (ReTAG)** | **78.09±02.57** | **64.30±02.21** |

- The performance drops when removing FCL, as eliminating 2-cells via fundamental cycles reduces the model to a purely edge-based graph. This shows that high-dimensional cycles are crucial for capturing richer structural context and stronger structural reasoning.

- The model's performance suffers when MTMP is discarded. Without hierarchical propagation, information exchange is confined to the 1-skeleton, preventing nodes and edges from exploiting higher-dimensional cycles. This shows that MTMP is essential for integrating multi-dimensional context and capturing topological dependencies.

- The performance of the model drops when removing CTCL. Without contrastive regularization over 2-cell representations, the model fails to enforce consistency among high-dimensional structures, leading to less discriminative embeddings. This shows that CTCL is essential for improving robustness and generalization in capturing cellular topology.

## 6 CONCLUSION

In this work, we explore the role of higher-dimensional topological structures in retrieval-augmented graph learning. We propose a novel framework, namely Retrieved Cellular Topologies-Augmented Graph Learning (ReTAG), which lifts input graphs into cellular complexes and constructs a knowledge base of multi-dimensional topology-aware subgraphs, termed cellular topologies. During inference, ReTAG retrieves structurally aligned cellular topologies and incorporates them via a multi-dimensional message-passing mechanism that captures complex topological dependencies beyond conventional pairwise relations. Furthermore, a cellular topological contrastive learning module is introduced to reinforce high-dimensional structural semantics by aligning features within individual topological cells. Extensive experiments across multiple graph tasks demonstrate that ReTAG outperforms state-of-the-art methods.

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

## A  NOTIONS

The notations in this paper are summarized in Table 4.

Table 4: Notations Tables in ReTAG.

| Notation | Definition |
|---|---|
| $\mathcal{G}$ | Graph space, consisting of graphs with nodes and edges. |
| $X$ | Regular cell complex space, consisting of a collection of cells. |
| $X^{(0)}/X^{(1)}/X^{(2)}$ | 0-, 1-, and 2-skeletons of $X$. |
| $x_\alpha$ | A cell in the cell complex $X$. |
| $\overline{x_\alpha}$ | Topological closure of cell $x_\alpha$. |
| $x^1 \prec x^2$ | Cell $x^1$ is a face of $x^2$ (i.e., $x^1 \subseteq \overline{x^2}$). |
| $\mathcal{X} = \{x_\alpha\}$ | Set of cells in $X$, indexed by $\alpha$. |
| $f : \mathcal{G} \to X$ | Cellular lifting map that maps graphs in $\mathcal{G}$ to regular cell complexes in $X$. |
| $I(x)$ | Importance score of a 0-cell complex $x$. |
| $p(x_i)$ | Probability of sampling 0-cell complex $x_i$. |
| $\varepsilon$ | Small constant used to ensure numerical stability in sampling probabilities. |
| $X_\sigma$ | Local cell complex constructed from the $k$-hop neighborhood around the master 0-cell $x_m^0$. |
| $k_b$ | Key for cell complex $X_\sigma$. |
| $k_q$ | Query key. |
| $S_\tau(\tau_q, \tau_b)$ | Temporal similarity score between the query and base cells based on the time difference. |
| $\mathbf{w}$ | Weight vector for combining temporal, 0-cell, and 2-cell similarities. |
| $w_1, w_2, w_3$ | Hyperparameter weights for temporal, 0-cell, and 2-cell similarity contributions. |
| $o_i$ | Task-specific output vector for the $i$-th cell in the complex $X_\sigma$. |
| $h_i$ | Embedding of the $i$-th cell in the complex $X_\sigma$. |
| $\boldsymbol{h}_m^0$ | Embedding of the master 0-cell in the complex $X_\sigma$. |
| $\boldsymbol{h}_c^0$ | Embedding of the center 0-cell in the query complex $X_q$. |
| $\boldsymbol{h}_q^2$ | Pooled two-dimensional topological feature for the query complex $X_q$. |
| $\boldsymbol{h}_\sigma^2$ | Pooled two-dimensional topological feature for the base complex $X_\sigma$. |
| $\mathcal{L}_{\text{CTCL}}$ | Cellular Topological Contrastive Learning (CTCL) loss. |
| $\tilde{\boldsymbol{h}}_{k,v}$ | Representation of 2-cell $x_k^2$ for the $v$-th view, aggregated from 0-cell subsets. |
| $\mathcal{L}_{\text{cls}}$ | Standard classification loss for supervised learning. |
| $\mathcal{L}_{\text{total}}$ | Total loss function combining classification and CTCL losses. |
| $\lambda$ | Regularization hyperparameter controlling the weight of the CTCL loss. |

## B  MORE MOTIVATION DETAILS

In this section, we provide more in-depth motivation for the proposed approach and elaborate on the challenges addressed by our method. We focus on how retrieval-augmented learning and higher-dimensional topological representations can help overcome some key limitations in traditional graph learning techniques.

### B.1  LIMITATIONS OF TRADITIONAL GRAPH LEARNING MODELS

Graph neural networks (GNNs) have made significant progress in learning graph representationsKipf & Welling (2017a); Hamilton et al. (2017c); Veličković et al. (2018), but they still face lim-

itations in capturing the complex and rich topological structure present in many real-world graphs. Existing methods predominantly rely on node-level and edge-level information for learning graph embeddings. However, these models often struggle to capture higher-dimension dependencies such as cycles, motifs, and other complex topological patternsBenson et al. (2016). This is especially problematic when dealing with tasks that require a deeper understanding of graph structure, such as graph retrieval and node classification in graphs with intricate structures.

While recent advancements in retrieval-augmented models (RAG) have shown promise in addressing these limitations, they primarily focus on leveraging node-level features and semantic retrieval. The rich topological semantics inherent in graphs are often underutilized, limiting the potential of these methods.

### B.2 WHY TOPOLOGICAL REPRESENTATIONS MATTER?

We argue that topological structures such as cycles, 2-cells, and higher-dimensional features play a critical role in enhancing graph learningHensel et al. (2021). Traditional graph representations, such as node embeddings or edge features, often overlook these higher-dimensional interactions, which are essential in many applications like social network analysis, molecular graph modeling, and knowledge graph explorationGilmer et al. (2017a). By lifting graphs into higher-dimensional cellular complexes, our method captures the intricate dependencies between nodes and edges that are often missed by conventional approaches.

In this work, we introduce ReTAG, a method that integrates both semantic and topological alignment to capture complex structural semantics within graphs. This enables the model to learn richer, more informative representations of graphs by considering both the node embeddings and the higher-dimensional topological interactions between nodes, edges, and other graph components.

### B.3 ADDRESSING THE LONG-TAIL KNOWLEDGE CHALLENGE

A distinctive challenge in graph learning is the long-tail distribution of knowledge. Unlike tabular or text data, graphs naturally follow a power-law degree distribution: a small number of high-degree hubs dominate the connectivity, while the vast majority of nodes have very low degree(Barabási & Albert, 1999; Newman, 2003). In retrieval-augmented settings, this imbalance is amplified— hub nodes, encoding frequent or common knowledge, are repeatedly retrieved and overrepresented, whereas long-tail nodes remain underutilized(Lewis et al., 2020; Borgeaud et al., 2022). This bias hinders the model's ability to capture rare but critical patterns.

The consequences of neglecting the long-tail are significant. In scientific knowledge graphs, rare compounds or niche experimental findings often drive new discoveries. In recommendation systems, cold-start users and infrequent items typically reside in the long-tail, and failure to model them exacerbates personalization gaps(Park & Tuzhilin, 2008; Celma, 2010). In graph classification, rare motifs (e.g., higher-dimension cycles) may carry essential discriminative signals. Thus, effectively incorporating long-tail knowledge is crucial for robust generalization.

To address this, we propose an **inverse importance sampling** strategy that rebalances the node distribution. Specifically, we define a node importance score $I(x)$ that combines centrality and degree, and assign sampling probability inversely proportional to it:

$$p(x) \propto \frac{1}{I(x) + \varepsilon}, \tag{18}$$

where $I(x)$ denotes a node importance score defined in the main text, $\alpha$ controls the balance, and $\varepsilon$ is a small constant for stability.

This reweighting reduces the dominance of hubs while amplifying the contribution of long-tail nodes. The trade-off parameter $\alpha$ ensures that global connectivity patterns remain preserved. As a result, the model learns to integrate both common and rare knowledge, achieving better generalization across diverse graph tasks, including cold-start recommendation and few-shot classification.

### B.4 THE ROLE OF MULTI-DIMENSIONAL TOPOLOGICAL MESSAGE PASSING (MTMP)

A key innovation in our method is the use of **Multi-Dimensional Topological Message Passing** (MTMP), which extends traditional message-passing schemes to higher-dimensional structures. While conventional GNNs propagate information between neighboring nodes(Gilmer et al., 2017b; Kipf & Welling, 2017b), MTMP allows information to flow across different topological dimensions such as cycles, motifs, and higher-dimension subgraphs(Benson et al., 2016; Bodnar et al., 2021a). This enables the model to learn more complex dependencies and capture the structural semantics that are essential for tasks like graph retrieval, classification, and link prediction.

MTMP is designed to operate over cellular complexes(Bodnar et al., 2021c; Hensel et al., 2021), which include not just nodes and edges, but also higher-dimensional features (2-cells, 3-cells, etc.). By using MTMP, we can propagate information across these higher-dimensional entities, allowing the model to understand the graph in a richer, more comprehensive way.

### B.5 CONTRASTIVE LEARNING FOR STRUCTURAL CONSISTENCY

To further enhance the learned representations, we introduce **Contrastive Topological Contrastive Learning (CTCL)** You et al. (2020b). The goal of CTCL is to enforce consistency in the learned embeddings of topologically similar structures. For example, if two nodes or subgraphs exhibit similar topological properties, their embeddings should be close in the learned representation space Chen et al. (2020). To achieve this, we adopt an *InfoNCE-style objective* van den Oord et al. (2018), which encourages embeddings of positive pairs (topologically consistent structures) to be similar while pushing apart negative pairs. This regularization mechanism improves the model's ability to generalize across graph structures with varying topologies.

### B.6 COMPARISON WITH EXISTING METHODS

Graph Neural Networks (GNNs) have garnered significant attention in both academic and industry communities for their robust capability to model complex, real-world data in various domains such as societal networks, biochemistry, and traffic systems Fang et al. (2025a; 2023); Gao et al. (2023); Li et al. (2022). By utilizing a message-passing mechanism, GNNs have gone beyond traditional node embedding approaches, enabling the capture of intricate relationships within graph-structured data. Despite their success, GNNs face significant challenges when generalizing across different graph modalities, domains, and tasks. This lack of generalization remains a major research frontier, especially when the graph structures differ substantially across scenarios Asai et al. (2023; 2024); Izacard et al..

While GNNs are capable of handling node-level features and local graph relationships, they often struggle with graphs that exhibit complex topological diversity or when graph structures vary significantly from the training data. This limits their ability to generalize effectively, particularly when tasked with complex graph patterns or when encountering graphs with unseen structures.

RAGraph improves upon this by integrating retrieval-augmented learning Jiang et al. (2024). It enhances generalization by retrieving semantically similar subgraphs from external sources and incorporating them into the learning process. However, RAGraph still focuses primarily on low-dimensional structures (nodes and edges), and while it introduces a retrieval mechanism to augment the model's contextual knowledge, it still lacks the ability to capture the more intricate, higher-dimensional topological features that are essential for tasks involving complex relational patterns. This is especially problematic when graphs involve structures like cycles or multi-dimensional topological motifs that are critical for accurate inference and prediction.

ReTAG, in contrast, builds on the retrieval-augmented approach but introduces topological alignment alongside semantic alignment. By utilizing cellular complexes Hajij et al. (2020), ReTAG captures richer topological features, such as cycles and loops, which are critical for reasoning over complex graph structures. Additionally, ReTAG combines retrieval with generation-augmented learning, allowing it to retrieve and generate relevant subgraphs during inference. This combination improves its generalization to unseen graphs and dynamic graph structures, making ReTAG more flexible and effective for tasks that require handling complex graph topologies.

In summary, while RAGraph and traditional GNNs improve graph learning by integrating retrieval mechanisms, ReTAG distinguishes itself by incorporating topological reasoning and generation-augmented learning, allowing it to better handle high-dimensional graph structures and enhance its performance on tasks requiring complex topological reasoning.

## C ADDITIONAL RELATED WORK

**Topological Deep Learning.** Topological deep learning (TDL) builds upon early advances in Topological Signal Processing (TSP) Barbarossa & Sardellitti (2020); Schaub et al. (2021); Rodden-berry et al. (2022); Sardellitti et al. (2021), which highlighted the importance of modeling higher-order relations beyond pairwise connections. This line of work has motivated the generalization of classical graph-theoretic tools to richer topological domains, such as simplicial complexes (SCs) and cell complexes (CWs). In particular, extensions of the Weisfeiler–Lehman test to SCs and CWs Bodnar et al. (2021d;b) have laid the theoretical foundations for higher-order message passing. Sub-sequently, a variety of neural architectures have been proposed, including convolutional designs for SCs and CWs Ebli et al. (2020); Yang et al. (2022); Hajij et al. (2020); Yang & Isufi (2023); Roddenberry et al. (2021); Hajij et al. (2022) and attentional formulations Goh et al.; Giusti et al. (2023). To unify these efforts, the combinatorial complex (CC) framework Hajij et al. (2023) was introduced, providing a general message-passing paradigm that encompasses SCs, CWs, and hyper-graphs. Parallel to these developments, Sheaf Neural Networks (SNNs) Hansen & Ghrist (2019); Hansen & Gebhart (2020); Bodnar et al. (2022); Battiloro et al. (2023; 2024); Barbero et al. (2022) have demonstrated the effectiveness of sheaf-based modeling in handling heterophily and capturing localized topological constraints.

While these works substantially extend graph representation learning to higher-dimensional do-mains, they largely remain confined to direct message passing within predefined complexes. Such approaches often exhibit limited generalization to unseen graphs with substantially different topolo-gies, a limitation that poses significant challenges for real-world applications where structural vari-ability is the norm. In contrast, our work goes beyond intra-complex learning by *retrieving and integrating multi-dimensional cellular topologies* from external corpora, enabling topology-aware retrieval and multi-level reasoning that enhance adaptability across diverse graph scenarios.

## D PROOFS

### D.1 PROOFS REGARDING PROPOSITION 1

*Proof.* **Contractibility of the spanning tree.** A spanning tree $\mathcal{T}$ is connected and acyclic, hence it is contractible. In topological terms, a contractible subspace can be continuously shrunk to a point within itself.

**Collapsing a contractible subspace.** Consider the quotient map $\gamma : G \to G/\mathcal{T}$ that identifies all points of $\mathcal{T}$ to a single vertex $v_0$. Collapsing a contractible subspace is a deformation retraction up to homotopy: there exists a continuous map $r : G \to G/\mathcal{T}$ and a homotopy $H : G \times [0, 1] \to G$ such that $H(x, 0) = x$ and $H(x, 1) = r(x)$ for all $x \in G$, with $r \circ \gamma \simeq \mathrm{id}_{G/\mathcal{T}}$. Therefore, $G$ and $G/\mathcal{T}$ are homotopy-equivalent.

**Induced isomorphism on first homology.** Homotopy-equivalent spaces have isomorphic homology groups. Hence the induced map $\gamma_* : H_1(G; \mathbb{Z}) \to H_1(G/\mathcal{T}; \mathbb{Z})$ is an isomorphism.

**Intuition in graph terms.** The spanning tree $\mathcal{T}$ contains no cycles, so collapsing it does not remove or merge any cycles in $G$. Each fundamental cycle in $G$ (formed by a non-tree edge and the unique tree path connecting its endpoints) is preserved in $G/\mathcal{T}$ as a loop based at the collapsed tree vertex. Therefore, the first homology group $H_1(G)$ — which measures independent cycles — remains unchanged. □

### D.2 PROOFS REGARDING PROPOSITION 2

*Proof.* Let $G = (V, E)$ be a finite connected graph and $\mathcal{T} \subset G$ a spanning tree.

**Existence and uniqueness of fundamental cycles.** For each non-tree edge $e = (u, v) \in E \setminus \mathcal{T}$, there exists a unique simple path $P_{\mathcal{T}}(u, v)$ in $\mathcal{T}$ connecting $u$ and $v$, by acyclicity of $\mathcal{T}$. Concatenating $e$ with $P_{\mathcal{T}}(u, v)$ defines a unique simple cycle

$$C_e = e \cup P_{\mathcal{T}}(u, v) \subset G.$$

Under the quotient map $\gamma : G \to G/\mathcal{T}$ that collapses $\mathcal{T}$ to a point, the tree-path $P_{\mathcal{T}}(u, v)$ is mapped to that point, so $C_e$ becomes a nontrivial loop in $G/\mathcal{T}$.

**Spanning and independence in homology.** The cyclomatic number of $G$ is $\beta_1(G) = |E| - |V| + 1$, which equals the number of non-tree edges. Thus there are exactly $\beta_1(G)$ fundamental cycles.

Any cycle in $G$ can be expressed as a linear combination of these fundamental cycles: traversing a cycle, each time a non-tree edge $e$ is encountered, the corresponding $C_e$ can be used to eliminate segments along $\mathcal{T}$.

These cycles are independent in $H_1(G)$, because under $\gamma$, each fundamental cycle maps to a distinct loop in $G/\mathcal{T}$, and loops around different edges of a wedge of circles are linearly independent in homology.

Hence the set

$$\{[C_e] \mid e \in E \setminus \mathcal{T}\}$$

forms a basis of $H_1(G)$.

**Topological summary.** Different choices of spanning tree yield different sets of fundamental cycles as edge sets, but the corresponding homology classes always span $H_1(G)$. Therefore, these loops capture all independent cyclic dependencies of $G$, providing a concise topological summary suitable for lifting $G$ into a higher-dimensional cell complex. $\square$

# E   MORE EXPERIMENT DETAILS

## E.1   DATASETS STATISTICS

### E.1.1   STATIC DATASETS

- **PROTEINS** Borgwardt et al. (2005): This dataset consists of protein graphs, where each node represents a secondary structure and each edge represents a relationship between amino acids or 3D space. The dataset contains 1,113 graphs with an average of 39.06 nodes and 72.82 edges per graph, with a density of 4.8e-2. This dataset is used for both node and graph classification tasks.
- **COX2** Rossi & Ahmed (2015b): A molecular structure dataset of 467 cyclooxygenase-2 inhibitors. Each node represents an atom, and each edge signifies a chemical bond (single, double, triple, or aromatic). The dataset is used for graph classification tasks, with each graph having an average of 41.22 nodes and 43.45 edges, and a density of 2.6e-2.
- **ENZYMES** Wang et al. (2022): A dataset of 600 enzymes, labeled into 6 categories according to their top-level enzyme classification. It contains 600 graphs with an average of 32.63 nodes and 62.14 edges, with a density of 5.9e-2. This dataset is used for both node and graph classification tasks.
- **BZR** Rossi & Ahmed (2015a): A dataset consisting of 405 ligands for the benzodiazepine receptor. The graphs represent each ligand, categorized into two groups. Each graph has an average of 35.75 nodes and 38.36 edges, with a density of 3.0e-2. This dataset is used for graph classification tasks.

### E.1.2   DYNAMIC DATASETS

We also use three publicly available datasets for dynamic recommendation (link prediction) tasks:

- **TAOBAO** : A dataset capturing implicit feedback data from Taobao.com, collected over 10 days. It is used for edge classification tasks, containing 204,168 nodes and 8,795,404 edges, with a density of 8.6e-4.
- **KOUBEI** : A dataset from Koubei, capturing 9 weeks of user interactions with nearby stores. It contains 221,366 nodes and 3,986,609 edges, with a density of 3.3e-4, used for edge classification tasks.

Table 5: Statistics of the experimental datasets.

| Statistics | Dynamic Graphs | | Static Graphs | | | |
| --- | --- | --- | --- | --- | --- | --- |
| | KOUBEI | AMAZON | PROTEINS | COX2 | ENZYMES | BZR |
| Nodes / Graph | 221,366 | 238,735 | 39.06 | 41.22 | 32.63 | 35.75 |
| Edges / Graph | 3,986,609 | 876,237 | 72.82 | 43.45 | 62.14 | 38.36 |
| Density | 3.3e-4 | 6.2e-5 | 4.8e-2 | 2.6e-2 | 5.9e-2 | 3.0e-2 |
| Graphs | 1 | 1 | 1,113 | 467 | 600 | 405 |
| Graph Classes | – | – | 2 | 2 | 6 | 2 |
| Node Features | – | – | 3 | 1 | 18 | 3 |
| Task | Edge | Edge | Node,Graph | Graph | Node,Graph | Graph |

- **AMAZON** : A dataset of product reviews from Amazon, spanning 13 weeks. It contains 238,735 nodes and 876,237 edges, with a density of 6.2e-5, used for edge classification tasks.

These datasets' detailed statistics are summarized in Table 5. The "Task" column provides information about the type of downstream task conducted on each dataset: "Node" denotes node classification tasks, "Graph" signifies graph classification tasks, and "Edge" indicates tasks related to link prediction. The "Type" column indicates whether the dataset is dynamic or static. For dynamic datasets, the "Snapshot Granularity" denotes the time granularity for each dataset. In our experimental setup, dynamic graphs are partitioned according to snapshots, while static graphs are partitioned either by node or by the entire graph.

### E.2 EVALUATION METRICS

#### E.2.1 NODE AND GRAPH CLASSIFICATION EVALUATION

For node and graph classification, we use prediction accuracy to measure the model performance.

#### E.2.2 LINK PREDICTION EVALUATION

For link prediction, we evaluate the recall and ranking quality of the recommendation effects following previous studies Yu et al. (2022a); He et al. (2020b). We use Recall@k and NDCG@k as evaluation metrics. Note that this task is a binary task. We denote the top-$k$ largest value as $\text{rel}_{ij}$, where $j \in [1, k]$ for node $v_i$.

**Recall@k**  Recall@k measures the ratio of true positive links contained in the top $k$ predicted links for each node. It is computed as:

$$\text{Recall@k} = \frac{1}{n} \sum_{i=1}^{n} \sum_{j=1}^{k} \text{rel}_{ij} \cdot I(A[i:] > 0), \quad (19)$$

where $\text{rel}_{ij} = 1$ if the $j$-th predicted link for node $v_i$ exists, otherwise 0. $I(\cdot)$ is the indicator function, and if $A[i:] > 0$, then $I(A[i:] > 0) = 1$.

**NDCG@k (Normalized Discounted Cumulative Gain)**  NDCG@k is computed by normalizing DCG@k (Discounted Cumulative Gain), which accounts for the position of correctly predicted links. DCG@k is defined as:

$$\text{DCG@k} = \frac{1}{n} \sum_{i=1}^{n} \sum_{j=1}^{k} \frac{\text{rel}_{ij}}{\log_2(j+1)}. \quad (20)$$

### E.3 BASELINE DETAILS

In this section, we present the details of baselines.

Table 6: Baseline Code URLs of Github Repository

| Baseline | Type | Code Repo URL |
|----------|------|---------------|
| GCN | Static | `https://github.com/tkipf/gcn` |
| GraphSAGE | Static | `https://github.com/williamleif/GraphSAGE` |
| GAT | Static | `https://github.com/PetarV-/GAT` |
| GIN | Static | `https://github.com/weihua916/powerful-gnns` |
| LightGCN | Dynamic | `https://github.com/kuandeng/LightGCN` |
| SGL | Dynamic | `https://github.com/wujcan/SGL-Torch` |
| MixGCF | Dynamic | `https://github.com/Wu-Xi/SimGCL-MixGCF` |
| SimGCL | Dynamic | `https://github.com/Wu-Xi/SimGCL-MixGCF` |
| GraphPro | Dynamic, Static | `https://github.com/HKUDS/GraphPro` |
| GraphPrompt | Dynamic, Static | `https://github.com/Starlien95/GraphPrompt` |
| PRODIGY | Dynamic, Static | `https://github.com/snap-stanford/prodigy` |
| RAGraph | Dynamic, Static | `https://github.com/Artessay/RAGraph` |
| ProNoG | Dynamic, Static | `https://github.com/Jaygagaga/ProNoG` |

**GCN Kipf & Welling (2016a)** : GCN is an end-to-end learning framework for graph-structured data. It utilizes neighborhood aggregation to integrate structural information, which is particularly effective in node classification and graph classification tasks.

**GraphSAGE Hamilton et al. (2017a)** : GraphSAGE is a general and inductive framework that leverages node feature information (e.g., text attributes) to efficiently generate node embeddings for previously unseen data.

**GAT Veličković et al. (2018)** : GAT is a spatial domain method, which aggregates information through the attention-learned edge weights.

**GIN Xu et al. (2019a)** : GIN utilizes a multi-layer perceptron to sum the results of GNN and learns a parameter to control the residual connection.

**LightGCN He et al. (2020a)** : LightGCN learns user and item embeddings by linearly propagating them on the user-item interaction graph, and uses the weighted sum of the embeddings learned at all layers as the final embedding.

**SGL Wu et al. (2021a)** : SGL supplements the classical supervised task of recommendation with an auxiliary self-supervised task, which reinforces node representation learning via self-discrimination.

**MixGCF Huang et al. (2021)** : MixGCF generates synthetic negatives by aggregating embeddings from different layers of raw negatives' neighborhoods to perform collaborative filtering.

**SimGCL Yu et al. (2022b)** : SimGCL applies unsupervised contrastive learning to enhance representation learning, making it suitable for link prediction tasks. It is applied to dynamic graphs to test its adaptability and performance.

**GraphPro Yang et al. (2024a)** : GraphPro extends GraphPrompt by introducing spatial and temporal prompts tailored for dynamic graph learning, enhancing the ability to capture both structural and temporal relationships within graph data.

**GraphPrompt Liu et al. (2023a)** : GraphPrompt integrates pre-training and downstream tasks using a unified template approach and employs task-specific prompts to enhance sub-task learning, applicable to both dynamic and static graph contexts.

**PRODIGY Huang et al. (2023a)** : PRODIGY focuses on facilitating downstream tasks through in-context examples and learning from the $X \rightarrow Y$ paradigm. It is implemented to enhance learning in both dynamic and static graphs by leveraging contextual learning strategies.

**RAGraph Jiang et al. (2024)** : RAGraph is a general retrieval-augmented graph learning framework that integrates external graph knowledge into pre-trained GNNs. It builds a key-value toy graph library, retrieves similar toy graphs via multi-dimensional similarity, injects knowledge into query graphs through intra/inter-propagation, and supports multi-graph tasks (node classification, link prediction, etc.) on static/dynamic graphs with high performance even without fine-tuning.

**ProNoG Yu et al. (2025b)** : ProNoG is a pre-training and prompt learning framework for non-homophilic graphs. It uses non-homophily tasks (e.g., GraphCL) for pre-training to learn universal graph properties, and designs a condition-net for downstream adaptation. The condition-net generates node-specific prompts by reading multi-hop neighborhood features (weighted by node similarity) and conditioning on node non-homophilic patterns, then adjusts node embeddings via element-wise product for fine-grained node/graph classification, especially in few-shot scenarios.

### E.4 FURTHER STUDY

We further evaluate the few-shot performance of ReTAG against baselines (RAGraph, Vanilla, GAT, GIN, GCN, GraphSAGE) on four datasets (PROTEINS, BZR, ENZYMES, COX2), as shown in Figure 5. Accuracy increases consistently as the number of shots grows from 1 to 5, confirming that additional supervision benefits few-shot adaptation. More importantly, ReTAG consistently outperforms all baselines across datasets and shot settings.

On **PROTEINS**, ReTAG improves over RAGraph by an average of +5.0pp, with a maximum margin of +8.9pp at 2-shot. On **BZR**, it achieves +3.6pp on average, maintaining superiority at all shots. On **ENZYMES**, ReTAG secures an average gain of +3.6pp, demonstrating robustness under low-accuracy regimes. On **COX2**, it yields a stable +2.5pp improvement on average, peaking at +4.5pp at 3-shot. Overall, ReTAG delivers consistent gains of roughly +2–6pp over the best baseline across datasets and shot scenarios.

These improvements stem from retrieval-augmented topology reasoning: lifting graphs into higher-dimensional cell complexes enables retrieval based on both semantic and topological alignment, while MTMP and CTCL further enhance representation robustness. Consequently, ReTAG achieves both higher accuracy and more stable performance than competing methods in few-shot settings.

## F STATEMENT ON LLM USAGE

We acknowledge the use of large language models (LLMs) as general-purpose assistive tools. Specifically, LLMs were used for polishing and improving the clarity of writing. However, all research ideas, methodological designs, experiments, and analysis were conceived, implemented, and validated by the authors. The LLM did not generate novel research contributions nor play the role of a scientific collaborator.

## G STATEMENT ON ETHIC AND REPRODUCIBILITY

This work adheres to the ICLR Code of Ethics. Our study does not involve human subjects, personally identifiable information, or sensitive private data. All datasets are publicly available and commonly adopted in the literature. We have carefully considered potential societal impacts: the proposed methodology is intended to advance representation learning and retrieval in cell complexes, without foreseeable risks of misuse. To ensure fairness, we mitigate bias in node sampling via inverse importance strategies. No conflicts of interest or external sponsorship influenced the research. For reproducibility, all datasets are publicly accessible, preprocessing steps are described

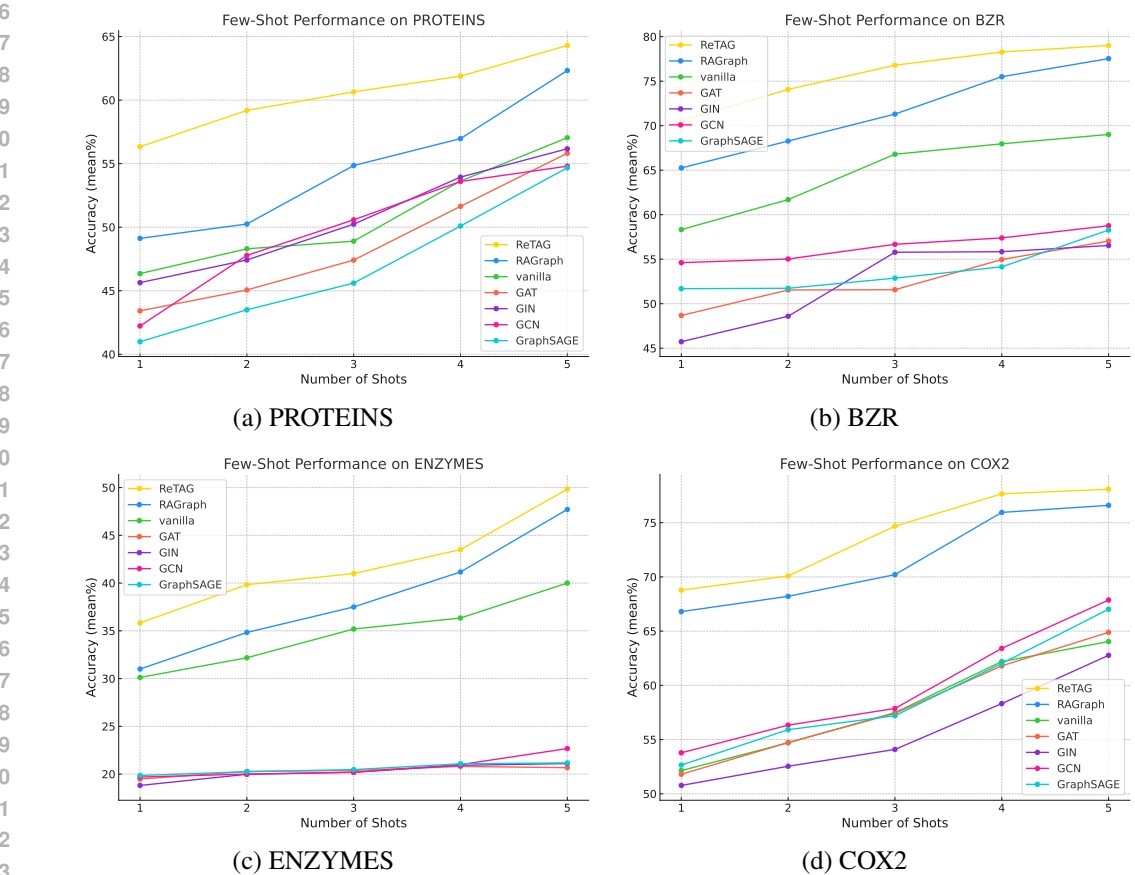

(a) PROTEINS

(b) BZR

(c) ENZYMES

(d) COX2

Figure 5: Few-shot accuracy comparison of **ReTAG** against baseline methods (RAGraph, vanilla, GAT, GIN, GCN, GraphSAGE) across four datasets (PROTEINS, BZR, ENZYMES, COX2). Results under 1–5 shots settings highlight the superior performance of ReTAG in few-shot learning scenarios.

in the supplementary material, and a self-contained code package is provided as part of the submission.

