# OpenReview forum: "ReTAG: A Retrieved Cellular Topologies-Augmented Graph Learning Framework"
_ICLR.cc/2026/Conference — ICLR 2026 Conference Withdrawn Submission_

### Official Review · Reviewer_KvS8 · 2025-10-27

**Soundness:** 2
**Presentation:** 3
**Contribution:** 2
**Rating:** 4
**Confidence:** 3

**Summary:**

This paper proposes ReTAG, a retrieval-augmented graph learning framework that leverages cellular complexes to capture higher-dimensional topological structures beyond nodes and edges. By retrieving and integrating multi-dimensional topology-aware subgraphs, ReTAG enhances reasoning over complex graph interactions. Experiments on multiple tasks show that ReTAG outperforms existing methods.

**Strengths:**

1. The paper tackles a cutting-edge problem in the graph learning field.
2. The main advantage lies in introducing high-dimensional topological structures (cellular complexes) as a topology-aware subgraph repository, enabling richer retrieval and reasoning beyond node- and edge-level information.
3. The paper is well structured and easy to follow, making the proposed ideas and technical contributions clear to the reader.

**Weaknesses:**

1. The explanation of how high-dimensional information contributes to the improvement is somewhat vague.
2. A discussion on the time complexity and efficiency of the proposed method is missing, which is important for understanding its practical scalability.
3. The experiments are conducted on relatively small datasets, leaving the performance on large-scale graphs unclear.

**Questions:**

1. The high-dimensional information in the paper appears to be mainly cycle motifs. However, graphs have diverse motif types, and the rationale for focusing solely on cycles is not discussed.
2. All experiments are conducted on protein and molecular datasets. It is unclear why social networks or other types of graphs are not evaluated, especially since the paper’s scope is not limited to biochemical domains.
3. The current datasets are relatively small; experiments on large-scale graphs are needed to assess the method’s scalability and practical performance.
4. A detailed analysis of the algorithm’s time complexity should be included to clarify its computational efficiency and scalability.

---

### Official Review · Reviewer_rEZU · 2025-11-01

**Soundness:** 3
**Presentation:** 3
**Contribution:** 3
**Rating:** 2
**Confidence:** 4

**Summary:**

Current RAG-based graph learning methods mainly focus on low-dimensional elements， which neglect the higher-dimensional structures in the graph.
To address this, this work proposes ReTAG, a framework that leverages cellular complexes to model and retrieve multi-dimensional topology-aware subgraphs.
Experiments on three different tasks demonstrate that capturing high-dimensinal elements can help improve model's performance.

**Strengths:**

1. ReTAG combines cellular complexes with RAG-based graph learning for the first time. It expands the topological dimension of graph representation learning, enabling models to capture high-dimensional structural information, which is highly beneficial for improving the performance of RAG-based graph learning models.
2. This work provides rigorous theoretical proofs, offers good explanations for all the theories used in the paper.
3. This work includes sufficient experiments, covering static and dynamic graphs, multiple tasks, and various datasets, with results showing a certain improvement compared to the baseline. Additionally, extensive hyperparameter analysis and ablation experiments have also verified the robustness of the proposed method.

**Weaknesses:**

1. ReTAG may have low generality. As ReTAG relies on the presence of obvious cycle structures in the graph, and its effectiveness for graphs without significant high-dimensional topology has not been thoroughly explored.
2. This paper does not discuss the potential impacts when the dimension of cellular complexes further increases and the scale of graphs continues to expand. It is suggested to attempt extending ReTAG to verification on some ultra-large-scale graphs, such as the large-scale datasets on Open Graph Benchmark (OGB).
3. This paper has several similarities to RAGraph in terms of writing. Additionally, many parts that are not the main innovations of this paper (such as inverse importance sampling) are included in the main text. It is suggested that the contributions from other papers be placed in the appendix, while the main text should focus more on highlighting the key achievements of this work.

**Questions:**

1. The main contribution of ReTAG lies in capturing high-dimensional topological structures in graphs, such as cycles and 2-cells, to encode richer topological and semantic information. However, in the case of two dynamic recommendation datasets like KOUBEI and AMAZON, all the data forms a bipartite graph, where cycle structures do not exist. Why high-dimensional topological structures can still play a role in such scenarios?
2. What are the time and space complexities of ReTAG in the processes of cellular complex construction and retrieval? Since constructing a cellular complex seems to be a highly complex process, will this significantly increase the time complexity of preprocessing and inference?

---

### Official Review · Reviewer_xKK3 · 2025-11-02

**Soundness:** 3
**Presentation:** 3
**Contribution:** 3
**Rating:** 6
**Confidence:** 3

**Summary:**

This paper proposes ReTAG, a retrieval-augmented graph learning framework that lifts graphs into cell complexes to retrieve and integrate higher-dimensional topological structures (e.g., cycles) during inference. By constructing a cell complex knowledge base, retrieving multi-dimensional cellular topologies, and employing multi-stage topological message passing plus cellular topological contrastive learning, the method aims to improve generalization under distribution shift. Experiments on node classification, graph classification, and link prediction demonstrate performance gains over modern GNN, prompt-based, and RAG-based baselines

**Strengths:**

1. The work bridges higher-order topological modeling and retrieval augmentation, addressing a gap in current RAG-based graph learning methods

2. Cell complex construction follows algebraic topology formalism (spanning tree contraction, 2-cells from fundamental cycles), improving conceptual rigor.

3. Consistently strong results across diverse graph settings with ablations validating the contribution of each component

**Weaknesses:**

1. Graph lifting and topological retrieval may be expensive at scale, raising concerns on the scalability of the proposed framework.

2. Many datasets are classical small benchmarks; it would strengthen claims to include larger heterogeneous data (e.g., OGB datasets, ZINC).

3. Important TDL baselines such as CW networks / simplicial networks are referenced but not reported experimentally.

**Questions:**

1. What is the practical computational overhead of cell lifting and topological retrieval during training and inference? Can the authors provide wall-clock measures and GPU memory usage relative to baselines (e.g., traditional GNN and CW network)?

2. CTCL and MTMP ablations are informative; could the authors also isolate: (1) retrieval without cellular lifting, (2) cellular lifting without retrieval, to more clearly separate the two contributions?

3. Can the framework naturally extend to higher-order cells beyond cycles (e.g., 3-cells in volumetric structures), and if so, what is required architecturally?

---

### Official Review · Reviewer_Bhm5 · 2025-11-03

**Soundness:** 2
**Presentation:** 2
**Contribution:** 2
**Rating:** 4
**Confidence:** 3

**Summary:**

This paper proposes a ReTAG (Retrieved Cellular Topologies-Augmented Graph Learning Framework) framework  that utilizes cellular complexes to capture multi-dimensional cellular subgraph information for effective graph representation learning. It retrieves the topological and semantic alignment of a given network and integrates them with a multi-dimensional message-passing mechanism in GNNs.

**Strengths:**

The method illustrates a beneficial way to integrate the topological and semantical information in networks.
The incorporation of contrastive learning helps in learning multi-domain information from the input graphs, thereby enhancing representation learning.
In an extensive experiment, the model shows better performance over other baselines.

**Weaknesses:**

The cellular-based model's learning seems complex, and there is no runtime or wall clock time given in the manuscript for the model. The authors did not explicitly explain
How does the model perform over different cyclic networks? Based on the presence of cycles, how does the model behave in final representation learning?
How do PageRank and degree-based information work regarding the model's performance? Could you please show the parameter analysis over the symbol $\alpha$
How does the value of $gamma$ impact the model? The parameter’s info is not provided with any statistics.
I think one mentioned dataset name, KOUBEI, should be BeiBei ([Zhu et el. 2021. CIKM])
Most of the datasets are about molecules, so how well this model works with molecular structures relies on how RAG shares molecular information from the knowledge base, which could improve the model's performance. Please provide some visualizations of the representations obtained through the RAG.
The number of datasets for node and graph classification is limited. Please increase the number of datasets for the experiment. For instance, you could consider using datasets [1] such as DD, NCI1, and PTR for graph classification.

 Morris, Christopher, et al. "Tudataset: A collection of benchmark datasets for learning with graphs." arXiv preprint arXiv:2007.08663 (2020).

**Questions:**

Several contrastive learning-based models dedicate themselves to node classification and graph classification tasks. Do you compare the framework with those models? Could you explain why your model is superior to the other models?
What are the statistics of cycles in the datasets? How does your model capture them?
Could you please provide some visualizations of the representations obtained from RAG? Could you please explain why graph representations created by ReTAG are considered superior to other methods?
How does high-dimensional/high-order graph learning maintain the time complexity? Specifically, how does high-dimensional/high-order graph learning maintain the time complexity on real-world datasets such as Beibei and Amazon? Please provide the whole clock time for training the model on these datasets.

---

### Note · Authors · 2025-11-26

I have read and agree with the venue's withdrawal policy on behalf of myself and my co-authors.